

# Quantifying uncertainty on sediment loads using bootstrap confidence intervals

Johanna I.F. Slaets[1], Hans-Peter Piepho[2], Petra Schmitter[3], Thomas Hilger[1], Georg Cadisch[1]

[1] Institute of Plant Production and Agroecology in the Tropics and Subtropics, University of Hohenheim, Garbenstrasse 13, 70599 Stuttgart, Germany

[2] Biostatistics Unit, Institute of Crop Science, University of Hohenheim, Fruwirthstrasse 23, 70599 Stuttgart, Germany

[3] The International Water Management Institute, Nile Basin and East Africa Office, Addis Ababa, Ethiopia

*Correspondence to*: Johanna I.F. Slaets, Institute of Plant Production and Agroecology in the Tropics and Subtropics, University of Hohenheim, Garbenstrasse 13, 70599 Stuttgart, Germany. (hanna.slaets@gmail.com)





**Abstract.** Load estimates are more informative than constituent concentrations alone, as they allow quantifying on- and off-site impacts of environmental processes concerning pollutants, nutrients and sediment, such as soil fertility loss, reservoir sedimentation and irrigation channel siltation. While statistical models used to predict constituent concentrations have been developed considerably over the last years, measures of uncertainty on constituent loads are rarely reported. Loads are the product of two predictions, constituent concentration and discharge, integrated over a time period, which makes it not straightforward to produce a standard error or a confidence interval. In this paper, a linear mixed model is used to estimate sediment concentrations. A bootstrap method is then developed that accounts for the uncertainty in the concentration and discharge predictions, allowing temporal correlation in the constituent data, and can be used when data transformations are required. The method was tested for a small watershed in Northwest Vietnam for the period 2010-2011. The results showed that confidence intervals were asymmetric, with the highest uncertainty on the upper limit, and that a load of 6262 Mg/year had a 95% confidence interval of [4331, 12267] in 2010 and a load of 5543 Mg had an interval of [3593, 8975] in 2011. Additionally, the approach demonstrated that direct estimates from the data were biased downwards compared to bootstrap median estimates. These results imply that constituent loads predicted from regression-type water quality models could frequently be underestimating sediment yields and their environmental impact.

**Key words:** constituent load, concentration prediction, discharge rating curve, sediment yield, resampling methods, Monte Carlo-method

## 1    Introduction

The environmental impact of processes such as erosion, sedimentation, eutrophication or degradation of aquatic ecosystems can only be quantified through reliable estimates of sediment, nutrient or pollutant loads (Walling and Webb, 1996). Monitoring constituent concentrations alone does not suffice as these provide information on in-stream quality, but offer no means to evaluate outcomes such as reservoir siltation, erosion, soil fertility loss and pollution at the watershed scale - both on- and off-site. Despite abundant literature developing appropriate procedures for load estimates, most studies do not report a measure of uncertainty on the load (Kulasova et al., 2012).

In this paper, we will use the example of one of the most commonly measured constituents, suspended sediment, but the methodology developed is applicable to any constituent load. For suspended sediment, the most frequently used method to estimate loads is the so-called rating curve method (Gao, 2008; Horowitz, 2008). In this approach, the suspended sediment concentration (SSC) is predicted by some form of least squares regression with (often log-transformed) discharge as the explanatory variable. This approach introduces two sources of uncertainty in the load equation: the uncertainty on the sediment concentration equation (the so-called sediment rating curve), and the uncertainty in the discharge, as discharge is usually not measured directly, but rather predicted from a regression with water level as predictor variable, the stage-



discharge rating curve. Any measure of uncertainty on the constituent load must take into account the uncertainty on both the constituent concentration, and the discharge.

Uncertainty of the sediment concentration prediction has been extensively discussed and, depending on the catchment characteristics, generally good concentration predictions are obtained with errors smaller than 15% (Horowitz, 2008). In some studies, however, the uncertainty is stated to be considerable. Smith and Croke (2005) for example reported that discharge only explained a quarter of the variability in the concentration data. Walling and Webb (1988) also found poor predictive power and suggested that seasonal differences of the relationship between discharge (Q) and constituent

concentration, non-simultaneity of Q and concentration peaks during storms, hysteresis and exhaustion effects are the most important causes of inaccuracy in concentration predictions. In their dataset, the sediment rating curve explained only 14% of total variance in the data and concentrations at the same level of discharge ranged over more than three orders of magnitude.

Furthermore, there are situations where the use of the rating curve method based on discharge is inherently unviable. For

example, in irrigated systems, the natural link between discharge and constituent load is disturbed (Slaets et al., 2014). As an alternative to the sediment rating curve method, turbidity sensors have often been employed - either as a substitute or in addition to discharge as predictor variable. Navratil et al. (2011) gave an overview of sources of uncertainty in sediment concentrations introduced by using turbidity sensors. They found that the most important ones were short-term turbidity signal fluctuations, the mechanism of the automatic water samplers and uncertainty on the discharge introduced by the fact

that discharge is not measured directly, but rather estimated from water level and a certain number of velocity measurements. Few studies to date have taken this error on the discharge rating curve into account explicitly when calculating uncertainty on load estimates (Vigiak and Bende-Michl, 2013; Rustomji and Wilkinson, 2008). Uncertainties of discharge, however, depend on several factors: the method chosen to estimate the discharge, site conditions and the time interval over which water levels are measured (Harmel *et al.*, 2009; Hamilton and Moore, 2012; McMillan et al., 2012; Tomkins, 2014).

Additional factors include errors in measuring the cross section, in determining the mean stream velocity, and field conditions during measurement such as change in stage, wind or ice obstruction (Sauer and Meyer, 1992). In general, discharge is estimated more accurately than constituent concentration. Discharge, however, enters the load equation twice – once as predictor variable for the concentration, and once multiplied with the concentration to get the instantaneous load. Therefore, it deserves further investigation whether or not the error on the discharge estimate can safely be ignored, and in

which circumstances. Finally, as discharge is frequently used as a predictor for constituent concentration, the two variables are correlated and their errors cannot be assumed to be independent. As a result, there is no textbook formula to estimate the variance of a constituent load.

Therefore, authors that do report a measure of uncertainty often select a method that is specifically geared towards the application of load estimation at hand, and not necessarily applicable to other sites, making it hard to compare results

throughout literature. Harmel et al. (2009), for example, developed a software tool to assess the errors introduced from



estimating discharge, sample collection, preservation and storage, and lab analysis. In this tool, each of these sources is considered to be the result of random variability following a normal distribution. The sources of error are assumed to be independent from each other and it is assumed that the errors follow an additive law, but these assumptions will not apply in all situations.

Moatar and Meybeck (2005) assessed uncertainty on nutrient loads by comparing loads based on a random subsample of measurement times, with a high-resolution load that is considered the "true" load. This approach is suitable for testing different methods and different temporal measurement resolutions of load estimation, but it does not assess the uncertainty of the "true" load, as it is assumed that with sufficiently high sampling frequency (in their case, daily), the measured load is equivalent to the actual load. More recently, two new candidate approaches have emerged to calculate confidence intervals

on loads, that have the potential to be generally applicable, regardless of the method used to calculate the load and the distributional assumptions made: bootstrap methods (Mailhot et al., 2008; Rustomji and Wilkinson, 2008; Vigiak and Bende-Michl, 2013) and Bayesian methods that result in credibility intervals (Pagendam et al., 2014; Vigiak and Bende-Michl, 2013).

The bootstrap is a Monte Carlo-type method, where a large number (B) of datasets are simulated – either by sampling with

replacement from the original data in the case of the non-parametric bootstrap, or by sampling from a fitted distribution in the case of the parametric bootstrap (Efron and Tibshirani, 1993). Bootstrap methods, however, were originally developed for independent, identically distributed random variables. In the context of sediment monitoring the assumption is that the observations used to build the regression model are independent in time. This can be realistic in fixed-interval sampling schemes where the sampling time interval is large, or in the case of discharge where measurements to build the stage-

discharge relationship are typically taken far apart in time. For sediment concentration, however, flow-proportional sampling is often performed to obtain samples at the highest concentrations. Those observations are usually taken closely together during storms and thus most likely are not independent in time (Slaets et al., 2014). In this case, it is necessary to use an adjusted version of the bootstrap that retains the serial correlation in the data intact (Lahiri, 2003). Such methods have already been explored in hydrology in relation to the discharge rating curve: Ebtehaj et al. (2010) and Selle and Hannah

(2010) uses block bootstrap methods to assess uncertainty on and improve robustness of model parameter estimates for discharge prediction.

To assess uncertainty on constituent loads estimated from continuous concentration and discharge predictions where serial correlation is present, we propose a bootstrap-based method to assess uncertainty on constituent loads that can be used with transformed data, that accounts for the uncertainty in both the sediment rating curve and the stage-discharge rating curve,

and that allows for serial correlation in the times series data. We checked if any of these requirements can safely be neglected in certain circumstances, and how they affect the resulting confidence intervals. Corresponding code in SAS was created and is available online to accommodate for these different scenarios (https://www.uni-hohenheim.de/bioinformatik/beratung/index.htm).



Our specific aims were: (i) to establish a generally applicable method to calculate confidence intervals on constituent loads, using bootstrap methods, (ii) to account for serial correlation in the data, (iii) to assess whether or not the effect of the uncertainty on discharge is negligible, (iv) to evaluate how data transformations affect the calculations, and (v) to determine the number of bootstrap replicates required to obtain reliable confidence intervals.

## 2 Material and methods

### 2.1 Discharge and sediment concentration

Discharge and suspended sediment concentrations were continuously monitored for a period of two years (1/1/2010 – 31/12/3011) in a small agricultural catchment in mountainous Northwest Vietnam. The catchment is located in the Chieng Khoi commune (21°7'60''N, 105°40'0''E, 350 m a.s.l), Yen Chau district, in the tropical monsoon belt where the rainy season begins in April and ends in October. Average annual precipitation is around 1200 mm, average annual temperature is 21 °C. The occurrence of typhoons is not uncommon especially at the end of the rainy season, and daily rainfall amounts can rise to 200 mm. The largest storm during the two years of this study was on the 12[th] of July 2011 and consisted of 73 mm of rainfall in three hours. The dominant soils are Alisols and Luvisols (Clemens et al., 2010). The landscape has an altitudinal range between 320 and 1600 m above sea level with slopes ranging from 0.05 to 65%. The measurement location is in a river at the outlet of a small watershed with a contributing area of 2 km$^2$, of which 0.6 km$^2$ consist of paddy fields. A 26.3 ha surface reservoir with a buffering capacity of 10$^6$ m$^3$ provides irrigation water for rice production via concrete irrigation channels, and the paddies drain into the monitored river. As the reservoir fills up with the progression of the rainy season, excess water is removed via a spillover which drains into the river, typically from July till October. During this period, discharge in the river is an order of magnitude larger than during times when the spillover is not active: mean daily discharge in the dry season equaled 0.08 m$^3$ s$^{-1}$(with a standard deviation of 0.13 m$^3$ s$^{-1}$), while mean daily discharge with the spillover active amounted to 1.22 m$^3$ s$^{-1}$ (with a standard deviation of 0.66 m$^3$ s$^{-1}$).

For the discharge monitoring, water levels were measured every two minutes for the river station using pressure sensors (EcoTech, Germany). The stage-discharge relationship was established with the velocity-area method (Herschy, 1995), where the velocity is measured with a propeller-type current meter (OTT, Germany) at one or more points in each vertical, depending on the water depth. The discharge is subsequently derived from the sum of the product of mean velocity, depth and width between verticals. Discharge measurements were never taken on the same day, and the closest time interval between two measurements was one week. The estimated discharge $\hat{Q}$ in m$^3$ s$^{-1}$ at time $i$ was then predicted from

$$log \hat{Q}_i = log\hat{\alpha} + \hat{\gamma} \log(h_i - \beta), \qquad (1)$$





where $h_i$ is the water level (in m) at time $i$, $\hat{\alpha}$ and $\hat{\gamma}$ the estimated rating curve constants and $\beta$ the measured sensor offset, with $\hat{\alpha}$ and $\hat{\gamma}$ estimated using the method of least squares on the log-transformed scale. This transformation was done to stabilize the variance.

As the irrigation management disturbed the natural relationship between Q and SSC, a turbidity-based method was used to monitor SSC. Turbidity was measured every two minutes with NEP395 sensors (McVan, Australia). 228 water samples were collected using a storm-based approach, by taking around twenty grab samples per sampled rainfall event in order to accomplish the best possible coverage of all concentration ranges. 188 storm flow samples were collected during 24 rainfall events over the two year duration of the study. Additionally, every two weeks a base-flow sample was taken. The sediment

concentration was determined gravimetrically on a sample of 500 ml by letting it settle overnight in refrigerated conditions, prior to siphoning off the supernatant and drying the remaining sediment at 35 °C, as is recommended for samples with very high sediment concentrations (ASTM, 2013).

Rainfall was quantified with a tipping-bucket rain gauge on a weather station (Campbell Scientific, USA). Events were defined based on rainfall data (no pause in precipitation for longer than 30 minutes) and lag times were added based on cross

correlation analysis as described in Schmitter *et al*. (2012). A total of 420 rainfall events took place and were monitored during the two year study period.

Continuous sediment concentrations were then obtained from a mixed model described in Slaets et al. (2014). The response variable, sediment concentration, was Box-Cox transformed to stabilize the variance using the SAS macro described in Piepho (2009). The optimal value of the transformation parameter was estimated by the maximum likelihood method, and

the selected value was the log-transformation. Predictor variables were chosen with forward selection based on Akaike Information Criterion (AIC). The model uses turbidity and discharge as quantitative predictor variables, and accounts for serial correlation. As surface reservoir irrigation management was present in the watershed, classic variables related to catchment characteristics such as hysteresis patterns and exhaustion effects were not suitable predictors of sediment concentration. The predictor variables turbidity and discharge were also log-transformed. All statistical analyses were

performed using the MIXED procedure of SAS 9.4, which can fit linear models with more than one random effect. The covariance structure used to model serial correlation in the present study was a first-order autoregressive (AR(1)) model, which was selected based on AIC. Assumptions of normality and homogeneity of variance were checked visually using diagnostic plots.

Conceptually, this concentration prediction error can also be separated into an underlying latent autoregressive process

generating the true concentrations, and an independently distributed measurement error corresponding to white noise in time series data. The white noise is equal to the error that would remain if two measurements were conducted at almost coinciding time points. This variability is typically attributable to measurement error and in spatial statistics, this is what is known as a nugget effect. In the MIXED procedure, this effect was fitted by using the local option in the repeated statement.





Validation was performed using five-fold cross validation, in which the dataset is split randomly into five parts, and each

part is used four times to calibrate the model, and one time for validation, so that each observation in the dataset is used for

validation once. Pearson's correlation coefficient (r) was calculated between the observed and predicted values resulting

from the validation. The SAS macro that performs k-fold cross validation for linear mixed models using the MIXED

procedure is described in Slaets et al. (2014). Additionally, event-based five-fold cross validation was performed, where all

samples belonging to single events were resampled jointly, rather than individual observations.

## 2.2   Bootstrap resampling procedure

The basic principle of the bootstrap is to mimic, as closely as possible, the original process of sampling from the population.

To this end, the bootstrap draws a large number (B) of random samples, of the same size as the original dataset, by sampling

observations with replacement from that original dataset – so that each observation can enter a bootstrap sample multiple

times. The sample statistic of interest (in this case, the sediment load) is then calculated for each of the B bootstrap samples,

resulting in B load estimates. The resulting empirical distribution of the sample statistic allows us to calculate measures of

uncertainty on the statistic, as B becomes large, but only if the bootstrap sampling mechanism is able to accurately reproduce

the original sampling process (Efron and Tibshirani, 1993).

As the key point of the bootstrap is to recreate the original sampling process, we need to understand the sampling processes

resulting in the annual sediment load. Since neither discharge nor constituent concentration are measured continuously,

annual loads are normally estimated by calculating the sum of instantaneous loads, measured at equally spaced discrete

points in time. The load at a time $i$ is then generated from

$$\hat{L}_i = \hat{Q}_i \times \hat{C}_i, \qquad (2)$$

where $\hat{L}_i$ is the estimated instantaneous load at time $i$ in g/s, $\hat{Q}_i$ is the estimated discharge at time $i$ in m³/s and $\hat{C}_i$ is the

estimated concentration at time $i$ in g/m³. These instantaneous loads are multiplied by a time factor accounting for the

monitoring interval. In the present study, for example, the factor was 120 s, as measurements were done every 2 minutes.

Monthly or annual loads in Mg can then be calculated by simply summing up the instantaneous loads for the whole time

interval and multiplying by a factor $10^{-6}$ to convert from mg to Mg:

$$\hat{L}_{1\,to\,t} = \sum_{i=1}^{t}(\hat{L}_i \times 120 \times 10^{-6}) \qquad (3)$$

Looking at Eq. (2) for the load estimate at a time $i$, there are really two separate sampling processes from two distinct

populations at work in the load estimation: firstly the sampling for the discharge rating curve (pairs of Q and h from the full

time series of Q and h pairs), and secondly, the samples used to build the sediment rating curve (observations of C and

hydrological predictor variables from the full time series of C, Q and turbidity). In order to assess the uncertainty of the

discharge equation, the bootstrap replicates can be created by simply sampling (Q, h)-pairs at random with replacement from

the original dataset. Simple random resampling assumes independence, which is dependent on the monitoring scheme: in the





case of our dataset, discharge measurements were never taken on the same day, and the smallest interval between two measurements was one week. In order to test this assumption, an AR(1) variance-covariance structure was fitted to the discharge data. As the AIC showed an increase of only two points compared to an independent structure, no serial correlation was present in the (Q, h)-pairs. Using Eq. (1), bootstrap discharges $\hat{Q}_i^*$ can be generated according to

$$log\hat{Q}_i^* = log\hat{\alpha}^* + \hat{\gamma}^*\log(h_i - \beta), \qquad (4)$$

where $h_i$ is the water level at time $i$, $\beta$ is the measured sensor offset, and $\hat{\alpha}^*$ and $\hat{\gamma}^*$ are the bootstrap estimates of the discharge rating curve parameters. These B predictions of Q, B being the number of bootstrap repetitions, must subsequently be fed into the bootstrapped sediment rating curve, as discharge typically is one of the predictor variables for SSC. But the previously described resampling mechanism cannot be applied to the observations used to build the sediment rating curve, as the simple random sampling assumes that the observations are independent. Water samples are often collected in a storm-based approach, as was done in this study, where they were collected sometimes only minutes apart during rainfall events. For this type of hydrological datasets where temporal autocorrelation is present, Ebtehaj et al. (2010) recommended the use of specialized sampling procedures that keep the serial correlation intact, such as the moving block bootstrap, the circular block bootstrap or the stationary block bootstrap, described in detail by Lahiri (2003) and applied to constituent concentrations by Hirsch et al. (2015).

Amongst these specialized methods, no clear winner has emerged up to now, and they require many choices - for example in terms of the block size - for which no general recommendation exists. But as the goal of the bootstrap is to mimic the original sampling process, there is an intuitive choice in the case of event-based sampling: the rainfall events form natural "blocks" or sampling unit, which is why water quality models used to predict continuous time series and thus new events should be validated on an event basis, rather than on a sample basis (Lessels and Bishop, 2013). So rather than sampling with replacement from the individual observations (water samples representing a single time point), all samples belonging to one event can be resampled with replacement, thus keeping all observations within one event together and maintaining the serial correlation intact.

On the other hand, base-flow samples are typically taken at fixed time intervals far apart in time (here every two weeks). They can therefore be considered to be independent and can be resampled by simple random sampling with replacement, thus bootstrapping individual water samples from single time points. An increase in AIC of only one point when fitting a first-order autoregressive covariance structure confirmed the lack of serial correlation in the base-flow samples. By resampling events with replacement for the storm flow samples and observations with replacement for the base flow samples, B time series of predicted sediment concentration $\hat{C}_i^*$ are generated from

$$\hat{C}_i^* = \hat{\delta}^* + (\hat{v}^*)^T X_i \,, \qquad (5)$$

with $\hat{\delta}^*$ and $\hat{v}^*$ the bootstrap estimates of the intercept and the regression coefficients, respectively, $(\hat{v}^*)^T$ the transpose of vector $\hat{v}^*$ and $X_i$ the design vector of the fixed effects. This is a generalized equation applicable to any linear model,



regardless of the number of predictor variables. If the sediment concentration was predicted using turbidity and discharge, for example, the bootstrap time series of predicted sediment concentration would be generated from

$$\hat{C}_i^* = \hat{\delta}^* + \hat{\eta}^* \hat{Q}_i^* + \hat{\kappa}^* T_i , \qquad (6)$$

where $\hat{\eta}^*$ is the bootstrap parameter estimate of the regression coefficient for $\hat{Q}_i^*$, the bootstrap discharge at time $i$ generated from Eq. (4), and $\hat{\kappa}^*$ the bootstrap parameter estimate of the regression coefficient for $T_i$, the turbidity at time $i$, respectively. This resampling process accounts for the uncertainty that arises from estimating the parameters of the sediment rating curve from a dataset with a limited number of observations. If there were an unlimited number of water samples available, the uncertainty of these parameter estimates would reduce to zero. But it is more realistic to assume that, even if there were a

very large number of samples available, there would still remain scatter in the real constituent concentrations around the equation, as the equation simply does not fully explain all the variation in sediment concentration. Sediment loads vary not only with discharge, but also with upstream sediment supply, which in turn depends additionally on geology, soil types, land cover and land use change or management, all influencing sediment quantity and quality (Walling, 1977). Therefore, there is a fundamental reason for the scatter in the data: sediment loads are inherently non-capacity loads. Even if there were an

unlimited number of samples available, this would not result in a perfect equation to predict sediment concentration. Therefore this additional uncertainty needs to be taken into account. For the discharge rating curve, if the river bed is stable and the stream bank vegetation does not change, the stage-discharge equation has a high accuracy and it is reasonable to assume the only error on the equation is measurement error, therefore this additional uncertainty is not a concern.

To introduce this second source of error on the sediment rating curve, Rustomji and Wilkinson (2008) and Vigiak and

Bende-Michl (2013) added an additional step to the bootstrap process: a randomly drawn residual from the original regression equation was added to the expected value of the constituent concentration, so that the predicted concentration included both the uncertainty of the parameters of the rating curve due to having a finite sample, and the uncertainty that arises from the fact that sediment concentrations simply cannot be perfectly predicted by any equation, regardless of how large the observed dataset would be. However, by randomly resampling from the residuals, it is assumed that these residuals

are independent.

When this assumption does not hold because samples are taken very closely together in time, as was the case for our dataset, the method can be modified so that the added errors reflect the temporal autocorrelation. To this end, the covariance parameter estimates from the original sample can be used as plug-in estimates. In the present dataset, an AR(1) structure was fitted to the data (Verbeke and Molenberghs, 2009), resulting in two covariance parameter estimates:  one for the

autocorrelation parameter ($\hat{\rho}$) and one for the residual error variance ($\hat{\sigma}_e^2$). The Restricted Maximum Likelihood algorithm was used to simultaneously estimate the fixed effects and the covariance structure (Patterson and Thompson, 1971). The use of the covariance parameter estimates obtained assuming a normal distribution of errors implies that the method is partly



parametric. This is necessary in order to take the serial correlation in the data into account. The bootstrap error term $e^*$ at time $i$ was then generated according to the following equation:

$$\hat{e}_i^* = \hat{\rho}^* \hat{e}_{i-1}^* + \sqrt{(1 - (\hat{\rho}^*)^2)} \times f_i^* \,, \tag{7}$$

where $\hat{\rho}^*$ is the bootstrap estimate of the autocorrelation parameter, and the error $f_i^*$ was randomly drawn from a normal distribution with mean zero and a bootstrap variance $(\hat{\sigma}_e^*)^2$. The bootstrap prediction of a sediment concentration at time $i$, including the error expected due to residual scatter in the data, is then given by

$$\hat{C}_i^* + \hat{e}_i^* = \hat{\delta}^* + (\hat{v}^*)^T X_i + \hat{\rho}^* \hat{e}_{i-1}^* + \sqrt{(1 - (\hat{\rho}^*)^2)} \times f_i^*. \tag{8}$$

In summary, the complete bootstrap process that accounts for uncertainty in the parameter estimates of both the discharge and sediment rating curves, and uncertainty due to residual scatter in the sediment concentrations consists of three steps (Figure 1):

(1) Resampling with replacement from the (Q, h)-pairs B times, in order to get B bootstrap stage-discharge equations; applying these equations to the continuous water level data to obtain B bootstrap time series ($Q^*$) for discharge;

(2) Block-bootstrapping the (C, turbidity, $Q^*$, rainfall) dataset by drawing whole events and base-flow samples with replacement, in each $i^{th}$ replicate plugging in the corresponding bootstrap $Q^*$ from Step 1, in order to get B bootstrap sediment rating curves; then applying these bootstrap sediment rating curves to the continuous turbidity, $Q^*$, and rainfall data to obtain B time series for the continuous suspended sediment concentration;

(3) Adding an error term to the concentration predictions to account for the residual scatter that is inherent to the sediment concentration.

In order to obtain a bootstrap estimate of the instantaneous load $L$ at time $i$, the equation is:

$$\hat{L}_i^* = \hat{Q}_i^* \times (\hat{C}_i^* + \hat{e}_i^*) \tag{9}$$

The residual scatter on the discharge is not added, as the stage-discharge rating curve has a much higher accuracy on the one hand, and on the other hand, velocity measurements are typically taken quite far apart in time, which would not allow modeling the serial correlation of the time series for discharge, but this would be needed because of the shortness of the time intervals considered here (2 minutes).

Finally, these bootstrap instantaneous load estimates can be summed up for the whole time interval, resulting in B estimates of monthly or annual loads:

$$\hat{L}_{1\ to\ t}^* = \sum_{i=1}^{t}(\hat{L}_i^* \times 120 \times 10^{-6}) \tag{10}$$

## 2.3 Data transformations

If the data are not normally distributed, it can be necessary to transform variables, as was done for this dataset with a Box-Cox transformation. In this case, the variables in question can simply be transformed before starting the bootstrap, and all the bootstrap estimates are obtained on the transformed scale. The back-transformation is then performed in Eq. (9) to obtain



load estimates on the original scale. For example, in a typical case where both discharge and sediment concentration need to

be log-transformed, the bootstrap predictions of discharge in Eq. (4) and of concentration in Eq. (8) will be on the log-scale. These predictors then need to be back-transformed to the original scale using the inverse of the logarithm.

This approach is applicable to any type of data transformation, and thus offers a flexible framework that can accommodate different methods of estimating the constituent concentration. However, if a modeled residual error term $\hat{e}_i^*$ is not included, care must be taken with the back-transformation. With nonlinear data transformations (the log-transformation and the Box-

Cox transformation being prime examples), predicted means cannot be naively back-transformed and interpreted as means on the original scale. Adding the modeled residual error removes the need to apply a correction factor that compensates for the underestimation of SSC that arises from doing the predictions on the transformed scale, as pointed out by Rustomji and Wilkinson (2008) and is therefore the recommended approach.

While discharge is also typically predicted on the log-transformed scale, in our dataset the variance was much smaller than

that of the concentration data. With a small variance, the log-normal distribution is nearly normal and therefore, the naïve back-transformation of $\log(\hat{Q}_i)$ should approximate the mean well.

### 2.4    Alternative option to simulate errors

If a data transformation is required and one does not want to explicitly simulate the residual scatter, then a correction factor must be applied to the back-transformed concentration. This correction is needed because the naïve back-transformation (for

example, taking the exponent of the predictions if the predictions are on the log-scale) does not yield a predicted mean, but rather a predicted median. While medians can be informative measures of central tendency for skewed datasets, they are not appropriate when the objective is to calculate a constituent load: loads are sums over equally-spaced time points, and in order to obtain an unbiased estimate of this sum over time intervals, we need to sum up estimates of the expected values, rather than the medians, for each interval.

The required correction factor is specific to the type of data transformation. For a logarithmic transformation, the expected value can be obtained by adding on half of the residual error variance to the predicted concentration on the log-scale before back-transforming. For other cases of the Box-Cox transformation, the correction depends on the selected transformation parameter. Solutions for specific examples of the transformation parameter can be found in Freeman and Modarres (2006). As the selected transformation in this dataset was the logarithm, the correction of adding half the residual error variance

before back-transforming was compared to the approach where the error is simulated, in order to see how this affects sediment load estimates and resulting confidence intervals.

### 2.5    Bootstrap confidence intervals



A straightforward way to calculate a confidence interval (CI) on a parameter after bootstrapping is the bootstrap percentile method (Efron and Tibshirani, 1993). If a 95% CI is required, the confidence interval would simply be calculated by ordering the bootstrap load estimates from small to large, and taking the 2.5[th] and the 97.5[th] percentile as the lower and the upper limit.

This method was used by Rustomji and Wilkinson (2008) on sediment loads and is transformation-respecting, also when the sample statistic is not normally distributed (Efron and Tibshirani, 1993). This property is important in the case of loads, because data are typically log-transformed. As a confidence interval depends on the tail of the empirical bootstrap distribution where fewer samples occur, a relatively large number of bootstrap replicates (upward of 500) are usually required to achieve acceptable accuracy (Efron and Tibshirani, 1993). How many exactly, depends on the statistic in question, and should be empirically tested for each case: when the process is repeated, the resulting CI should not greatly differ, otherwise the number is too small. In the present dataset, a choice of 2000 bootstrap replicates yielded replicable results.

Improving upon the bootstrap percentile method, Efron and Tibshirani (1993) proposed bias-corrected and accelerated intervals, used by Vigiak and Bende-Michl (2013). Unfortunately, this approach requires an even larger number of bootstrap replicates than the percentile method to sufficiently reduce the Monte Carlo sampling error. This is a disadvantage when working with hydrological time series, as the datasets typically contain a large number of records already. This method then quickly becomes time consuming, and therefore in this paper, preference was given to the more intuitive and less computationally intensive bootstrap percentile method.

## 2.6 Identifying hydrological drivers of uncertainty

The proposed three-step bootstrap process offers an opportunity to assess the importance of different aspects of the load calculation on the accuracy of the estimate. By leaving out step 1 (bootstrapping the Q-h pairs) and just using Q as predicted by the discharge rating curve from all observed data points, confidence intervals can be obtained that only take into account the uncertainty on the sediment rating curve. If the resulting confidence intervals closely resemble the confidence intervals calculated with the full approach, this would mean that the uncertainty on the sediment concentration is what drives the uncertainty on the loads, thus supporting that the error on the discharge is negligible compared with other sources of uncertainty (eg. Némery et al., 2013, Vigiak and Bende-Michl, 2013).

As the accuracy of the stage-discharge relationship depends on the type of streambed, the method chosen and the amount of measurements taken, this assumption might also hold true for some watersheds such as the one in this study, where the relationship had a high $R^2$, but not for others. To determine at which point the uncertainty on Q must be taken into account for the load confidence interval, data sets of (Q, h)-pairs were simulated with decreasing $R^2$ (0.95, 0.90, 0.85 and 0.80), and were each used as input dataset for bootstrapping the stage-discharge relationship (Step 1 in Figure 1) in order to test the





sensitivity of the confidence intervals to the accuracy of the discharge rating curve. The datasets with a fixed realized $R^2$

were simulated by a rescaling of errors which is described in Appendix A, and SAS code to perform the simulation can be found in the supplementary materials.

## 3   Results

### 3.1   Rating curves and load estimates

The coefficient of determination of the stage-discharge relationship was 0.98 (n=15, Figure 3). Homoscedasticity was

observed on the log-transformed scale (Figure 3). For the sediment rating curve, Pearson's r between observed and predicted values on the log-transformed scale was r=0.75 after five-fold cross validation (n=228, Figure 4). Event-based cross validation yielded very similar results, demonstrating the robustness of the model (r=0.77). Thus in the case of our dataset, the discharge rating curve explained a higher proportion of the variance than the sediment rating curve, as is typical. Again, homoscedasticity was observed on the log-transformed scale (Figure 5). The bootstrap parameter estimates for $\rho$ of the

AR(1) process varied from 0.56 to 0.93 with a mean of 0.77, showing the block bootstrap kept the serial correlation intact as required.

The size of the estimated load depended on the method chosen for estimation. First, the load was calculated directly from the model estimates based on the full datasets, without bootstrapping (Direct estimate in Table 1). The sediment concentrations in this case were back-transformed by applying the correction appropriate for log-transformed data, which is to add half the

residual error variance before back-transformation. Second, the median of the bootstrap estimates of the sediment load was taken, where, identically to the first case, the concentrations were corrected by adding half the residual error variance before back-transforming (Bootstrap without modeled error in Table 1). Third, the median of the bootstrap estimates was taken for the bootstrap process that included a modeled, autoregressive error term (Full bootstrap method in Table 1).

For this last estimation method, the annual sediment load was estimated to be 6262 Mg in 2010 and 5543 Mg in 2011 (Table

1). When the median from the bootstrap sediment load estimates was taken without modeled error, but rather applying the back-transformation correction, the load was approximately 5% higher for both annual and monthly load estimates (Table 1 and Figure 6). The annual loads thus amounted to 6575 Mg in 2010 and 5839 Mg in 2011. Finally, if sediment loads were estimated not by bootstrapping, but directly from the data, then the results were around 10% lower compared to the first estimates, at 5607 Mg and 4997 Mg, respectively, in 2010 and 2011.

In all three approaches the difference between the two years remained consistent and all estimates were within the bounds of the confidence intervals, both for those calculated by modeling error and those calculated by adding half the variance before back-transformation.

In this particular 2 km$^2$ catchment, the annual sediment export of 6262 and 5543 Mg cannot be interpreted as resulting in average erosion rates of approximately 30 Mg ha$^{-1}$ due to the irrigation management in the catchment. A large part of the





sediments are not eroded within the watershed but released from the irrigation reservoir, which has a contributing area of 490ha, either via the irrigation channels or through a spillover mechanism which releases excess water when the reservoirs maximum capacity is reached. In 2011, the former mechanism introduced around 800 Mg of sediments to the catchment, and the latter resulted in a load of 1556 Mg (Slaets *et al.*, 2015). True upland area erosion rates were estimated at 7.5 Mg ha$^{-1}$ a$^{-1}$ (Slaets *et al.*, 2015).

**3.2    Width of confidence intervals for sediment loads**

Before looking at the bootstrap confidence intervals, the histograms of the bootstrap load estimates were evaluated (Figure 7). The histogram of the 2000 bootstrap estimates looked reasonably smooth, so we concluded that sample size was adequate for the percentile bootstrap. For both years and both for the full method and the method without modeled error, the histograms were found to be skewed to the right, even when the loads were log-transformed. This skewedness means that, in

the case of our dataset, the assumption of normality would not hold for estimated annual loads.

As a result of the distribution of the loads, the confidence intervals were always asymmetric, with the difference between the upper limit and the estimate around 80% larger than the difference between the estimate and the lower limit. The width of the intervals – the difference between the upper and lower limit of the interval – , varied between years and between methods, while remaining in the same order of magnitude (Table 1). In 2010, the interval was always wider, regardless of

which method was chosen, for the annual as well as for the monthly loads (Table 1 and Figure 6). The year 2010 did contain a smaller proportion of the samples (73 out of 228) and this could be a cause for the difference. For the monthly loads (Figure 6), confidence intervals were widest during months with the highest sediment loads (July till October), when excess reservoir water gets exported via the river.

The bootstrap method affected the width of the confidence interval as well. The monthly and annual intervals resulting from

applying a back-transformation correction were consistently wider than those resulting from the bootstrap process that modeled the autocorrelated error: not modeling the error changed the interval (limits in Mg) from [4331, 12267] to [4372, 14586] in 2010 and from [3593, 8975] to [3713, 10410] in 2011 – in both cases an increase in width of about 20%. The change was due to an increase in the upper bound of the interval, while the lower limits remained very similar. These results show that performing the back-transformation correction is only a very rough method of adjusting the predicted

concentrations on the original scale, as this approach does not take the serial correlation in the data into account. For the monthly load estimates, the largest differences in confidence interval width between the full method and the back-transformation without simulated error were in July and August 2010, the months with the highest estimated loads (Figure 6).

**3.3    Hydrological drivers of uncertainty**



When, rather than applying the full bootstrap method, we did not bootstrap the discharge rating curve (meaning, we left out Step 1 of the process in Figure 1), the width of the confidence interval decreased, as one less source of error is taken into account. In 2010, this changed the CI from [4203, 11649] without accounting for uncertainty in the discharge rating curve to [4331, 12267] when accounting for this uncertainty on discharge; and from [3521, 8397] to [3593, 8975] in 2011 – including discharge therefore resulted in a respective increase in width of 6 and 9%. Similarly, in the monthly load estimates, not

bootstrapping the discharge resulted in confidence interval widths up to 37% smaller than those calculated with the full method (Figure 6). Months with low flow showed equally compressed confidence intervals as months with high discharge, during which the reservoir spillover was feeding the river (July till October).

The accuracy on the (Q, h)-relationship in this particular dataset was very high with an $R^2$ of 0.98. As not all monitoring programs can establish accurate discharge rating curves, the (Q, h)-dataset was replaced by a simulated dataset with

increasingly lower coefficient of determination to test how this further affects the uncertainty on the load estimate (Figure 8). While the width of the confidence interval keeps increasing with decreasing $R^2$, including the discharge also affects the confidence interval for a high $R^2$. In fact, changing from not bootstrapping the discharge ($R^2$=100% in Figure 8) to bootstrapping the real discharge dataset, which has an $R^2$ of 0.98, resulted in a 7% increase in width. On the other hand, the confidence intervals show little differences at an $R^2$ of 0.95, where the width was 9003 Mg, and at 0.90, when it reached up

to 8795 Mg, equivalent to an 11% increase in width. At a coefficient of determination of 0.85, the CI was 17% wider than the original CI and at 0.80 finally, the width increase reached 20%. As can be seen from Figure 8, the change in width was mainly due to an increase in the upper limit of the confidence interval. Hence, the lower limit decreased only slightly.

The bootstrap approach where the concentration prediction error was separated into an underlying latent autoregressive process generating the true concentrations, and an independently distributed measurement error corresponding to white noise

in time series data, did not converge for 906 out of 2000 runs. Convergence problems are very common when trying to fit nugget models as these models tend to be difficult to fit. Particularly AR(1) type error structures are prone to these issues, as there is an inherent confounding between parameters of the independent white noise component and the autocorrelated component (Piepho et al., 2015). In a bootstrap setting where convergence was already an issue, adding such an effect was not feasible in the case of our dataset. For exploratory purposes, the nugget can be fitted to the original dataset without

bootstrapping, in order to examine the contributions of the respective error components. Results of this exercise showed that indeed, the measurement error (0.67) was large compared to the latent process variance (0.09), the former being due to sensor error, both from the turbidity sensors and the pressure sensors for discharge, the manual grab sample process which may not accurately represent the mean concentration across the cross section, and laboratory error in determining the sediment concentrations. The error separation thus indicates that focusing on these factors could yield substantial

improvement in the sediment rating curve.

## 4    Discussion



### 4.1  Load estimates, data transformations and bias

As was shown in Table 1, the annual sediment load estimates differ depending on the method selected. While it is encouraging that all estimates are within the 95% confidence interval limits, choosing a different method can lead to
anything from an underestimation of 10% to an overestimation of 20% compared to the median of the full bootstrap process. Two issues play a role in these differences: the back-transformation of the sediment concentrations, and bias in the estimate of the annual load.

The effect of back-transforming the concentration predictions is visible when comparing the medians of the bootstrap estimates with and without modeled autocorrelated error. When the error was not modeled, the estimate itself increased by
around 5% in both years, corresponding to an absolute increase of around 300 Mg of sediment, and the CI became wider. Essentially, adding half the variance before back-transformation is a very rough way of estimating expected values of concentrations at original time points – as shown by the larger CI - because it does not take the serial correlation in the data into account. If the naïve back-transformation would be applied, without any variance correction, the resulting estimates would be even lower than those where we add half the variance before back-transformation: around 4200 Mg in 2010 and
3700 Mg in 2011, or an underestimation of approximately 2000 Mg.

While the latter is a relatively common approach to implement the back-transformation of constituent concentration predictions which are typically predicted on the log-scale, it may not be the most appropriate solution when the concentrations are used to calculate loads. The crucial issue with load calculations is that a load is a sum over time points, which is essentially the same as computing an arithmetic average, and for that we need to estimate the expected values for
the individual time intervals. If the predicted value on the log-scale is simply back-transformed, we are estimating medians of the concentrations, and while this may be appropriate if one is only interested in the concentrations, these medians cannot be multiplied with discharge and summed up to accurately predict a load.

When the bootstrap process includes a simulated, autocorrelated error, the result of that process is not a mean or a median concentration, but rather a simulated realization of an observed process. When it is not desired to simulate the error in the
bootstrap process, then applying a back-transformation correction is an alternative, but the confidence intervals should be expected to be wider, as adding on half the residual error variance before back-transformation ignores the serial correlation. An alternative back-transformation correction often used in literature, Duan's smearing correction, similarly assumes independent and identically distributed errors and is therefore not suitable for datasets where serial correlation is present (Duan, 1983). Duan's is a non-parametric correction where the sample average of the exponentiated residuals from the
model is used as the correction factor, in contrast with the two parametric approaches we used.

The back-transformation method of the concentration predictions, however, is not the only force at work: the direct estimate from the data and the bootstrap median without modeled error are quite far apart, even though they both use the same back-transformation correction. Statistics, unless they are very simple (for example a sample mean), will typically have some bias.



Bootstrapping can in fact be used to identify and correct bias even when the true underlying distribution is unknown; therefore in most cases the bootstrap estimate will typically be different as it removes this bias (Efron and Tibshirani, 1993). There are alternative methods in literature intended to remove the bias on load estimates (Ferguson, 1986), but as the correction will depend on the variance of the data, numerical corrections are not generally applicable. However, as one would need to bootstrap in any case in order to produce a CI of the load, taking the median of the bootstrap estimates is a straightforward way to obtain constituent load estimates.

Regarding the data transformation, while the sediment concentration was log-normally distributed, the log-transformed load estimates were not normally distributed (Figure 7, right panel). This non-log-normality of our loads does not affect the viability of the bootstrap approach, but it does limit the applicability of methods that use the log-normality assumption of the load to estimate a variance for the load, as was done for example by Wang et al. (2011) in an approach that used the delta-method as an alternative way to assess uncertainty on annual sediment load estimates.

## 4.2 Confidence interval width and model selection

The results showed that the CI's are relatively wide and asymmetrical with a much larger uncertainty on the upper limit. And comparing the two years, when the estimated load was higher, the uncertainty on it was larger as well (i.e. in 2010). This is a trend consistent with other studies (Kuhnert et al., 2012; Rustomji and Wilkinson, 2008). Although it is difficult to compare uncertainty calculated with other methods in different catchments, our confidence intervals are of the same order of magnitude as the CI's in those two studies. For example, Kuhnert et al. (2012) calculated 80% confidence intervals on an annual load of 5232 Mg (n=122), and the resulting limits were [3512, 7775]. In comparison, 80% confidence intervals for our dataset were [4186, 7403] for a load of 5543 Mg in 2011.

The factors governing the width of a confidence interval are essentially the sample size and the accuracy of the two rating curve estimates. If the sample size is large and the variation explained by the rating curves large, but the confidence intervals are very wide, one possible cause is that the concentration prediction model was over-fitted, resulting in a very high apparent percentage of variance explained by the model but a poor predictive power when the model is interpolated to the whole time series. This can be shown by just adding additional predictor variables to our selected model. If we add the variables 'water height in the reservoir', 'discharge irrigated to the paddy fields' and 'Julian day-of-year' (the last one both linear and quadratic) to the model, the percentage of variance explained increases from 58% to 71%. When this extended model was used to estimate the annual sediment load, however, the confidence interval was inflated by two orders of magnitude, resulting in a width of 5 564 076 Mg.

These effects on the CI indicate that indeed, overfitting is a concern even when interpolating within the time series. The risk of overfitting is particularly high with more complex models (Burnham and Anderson, 2002), as was demonstrated as well with the example above, and it is not uncommon in load estimations to fit models that are very flexible (eg. spline functions,





sigmoid functions) and / or have used a large number of predictor variables to a relatively small dataset. In such cases, bootstrap uncertainty assessment can be an additional tool both for model selection, and for evaluating model fit. The change in percentage variance explained is less pronounced after cross validation, and ranges from 56 to 64%, implying that the cross validation penalizes at least partially for any overfitting. Water quality models, however, are often not validated and only the $R^2$ resulting from calibration is reported, leaving readers no means of assessing over-parameterization of the model.

Studies with smaller datasets where more variables are included in the model should be particularly encouraged to report measures of uncertainty on load estimates. In the case of large datasets where a simple model, such as linear regression with one or two predictor variables is used, the variability in the data explained by the model resulting from calibration only is less likely to deviate strongly from the result of a validation.

An issue related to overfitting and model selection could also be seen in the selection of the variance-covariance structure. In

a previous study on water quality prediction models (Slaets et al., 2014), a power decay structure was selected based on AIC. The power model, however, gave convergence problems in some of the bootstrap replicates due to having fewer different observations to estimate the variance-covariance parameters. Based on these bootstrap results, the covariance structure that models serial correlation in this paper was changed from the previous power model to a simpler first-order autoregressive structure which did not have any convergence issues.

## 525    4.3    Bootstrapping discharge and error propagation

One would expect that, as the sediment rating curve has much more uncertainty than the discharge rating curve, excluding the latter would not affect the confidence intervals much, but for our dataset, this assumption did not hold: even having a discharge rating curve with high accuracy ($R^2$ of 0.98), its uncertainty had a considerable effect on the load estimate, which increased from 6389 Mg when not bootstrapping the discharge, to 6781 Mg when bootstrapping the discharge.

This result underscores the importance of error propagation in uncertainty assessments. Even though the discharge rating curve has a high accuracy, an estimate of Q is used as a predictor variable for concentration, and the concentration then gets multiplied with the estimate of Q, and so the effect is not as small as one would expect based on the $R^2$ of both rating curves. Krueger et al. (2009) similarly found discharge to contribute to uncertainty of sediment transfer. If we assume that most discharge rating curves have around 95% of explained variance, this could imply that most measures of uncertainty in literature

are too conservative by about 10% in terms of width of the CI and that this increase would be mostly on the upper limit of the interval – implying that minimum impact estimates would not be affected much, but that literature to date would underestimate worst case scenarios of sediment yield, nutrient loss or erosion with about 10%.

## 5    Conclusions



The approach developed in this paper provides a means to assess uncertainty on any type of constituent load, which was
calculated from continuous constituent concentration and discharge predictions estimated with regression-type methods.
Compared to Ordinary Least Squares regression methods to obtain load estimates, bootstrap estimates resulted in bias-
corrected estimates that can take serial correlation into account when present, as well as providing a measure of uncertainty
on the load estimate.

The results show that, even when the uncertainty of the discharge rating curve is small, it is important to take into account
that the errors propagate by using discharge both as a predictor variable for constituent concentration and in the
instantaneous load equation. Application of the method in different watersheds, at different spatial and temporal scales could
elucidate whether discharge is an important driver of uncertainty in those settings as well.

The confidence intervals resulting from our proposed method showed that the uncertainty on the loads is quite large and is
mostly on the upper limit of the estimate, as the intervals were strongly right-skewed. This asymmetry implies that, wherever
load estimates are used to assess environmental impact, without reporting an uncertainty assessment, the maximum impact
could be severely underestimated.

Additionally, the bootstrap process demonstrated that load estimates are biased downwards if calculated directly from data
with increasing variance that has been transformed. While some alternative bias corrections are available, these are not
consistently used, and this is another factor contributing to the underestimation of constituent loads thus far reported in
literature. Taking the median of the bootstrap estimates is an easy and generally applicable way to obtain unbiased estimates.
Reporting uncertainty is especially important when water quality models are complex. There has been a great increase in the
use of more complex predictive methods for water quality, for example the use of Artificial Neural Networks, Random
Forests or Generalized Additive Models (Berk, 2008). The advent of these methods makes the consistent reporting of
measures of uncertainty even more essential: the more complex a model is, the more prone it is to overfitting (Burnham and
Anderson, 2002), as was demonstrated by the inflated confidence intervals when adding predictor variables to the sediment
concentration model. Some measure of uncertainty should systematically be shown for any load estimate, and the method
developed in this paper provides a flexible framework to do so.

## 6   Data availability

The source code for the bootstrap analysis with the SAS software that was used for the load estimates and corresponding
confidence intervals is freely available at https://www.uni-hohenheim.de/bioinformatik/beratung/index.htm together with
necessary input files for testing. The full dataset is available from the authors upon request (hanna.slaets@gmail.com).

## 7   Acknowledgements





The field work data in this study was collected in the frame of the Collaborative Research Center "The Uplands Program", a DFG-funded project in collaboration with Tran Duc Vien at the Hanoi University of Agriculture. The authors gratefully acknowledge the work of field assistants Do Thi Hoan and Nguyen Duy Nhiem, and the laboratory analyses were done at the Central Water and Soil Lab of Hanoi University of Agriculture, under supervision of Nguyen Huu Thanh by Dang Thi Thanh Hue and Phan Linh. The source code for the bootstrap analysis done in this study using SAS is freely available at https://www.uni-hohenheim.de/bioinformatik/beratung/index.htm together with necessary input files. The SAS code used to simulate a dataset with a fixed realized $R^2$ can be found in the supplementary materials. The full dataset is available from the authors upon request (hanna.slaets@gmail.com).

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

## Appendix A

In order to obtain $(\log(Q), \log(h))$-pairs with a certain $R^2$ (eg. 0.95), we started by simulating a dataset of the same number of
observations of the original dataset. We thus obtained pairs of $\log(Q_i)$ and $\log(h_i)$ using the original discharge rating curve,



which is a regression model $y_i = \eta_i + e_i$, where $\eta_i = \alpha + \beta x_i$ and the errors were randomly drawn from a normal distribution which can have any variance, as the errors will be rescaled later.

Next, we computed the total sum of squares, $SS_y$, and the residual sum of squares, $SS_e$, for this simulated dataset. We subsequently replaced the simulated errors $e_i$ by re-scaled errors $e_i^* = c e_i$ and used these to compute re-scaled simulated data $y_i^* = \eta_i + e_i^*$. A scaling constant $c$ was chosen in such a way that the desired coefficient of determination results:

(A1) $R^2 = 1 - SS_e/SS_y$ and $R^{2*} = 1 - SS_e^*/SS_y^*$,

(A2) The residual error sum of squares is a quadratic form of errors only. It follows that

$$SS_e^* = c^2 SS_e$$

(A3)

$$SS_y^* = \sum_{i=1}^{n} (y_i^* - \bar{y}^*_{.})^2 = \sum_{i=1}^{n} (\eta_i + c e_i - \bar{\eta}_{.} - c\bar{e}_{.})^2 = \sum_{i=1}^{n} [(\eta_i - \bar{\eta}_{.}) + c(e_i - \bar{e}_{.})]^2$$

$$= \sum_{i=1}^{n} (\eta_i - \bar{\eta}_{.})^2 + 2c \sum_{i=1}^{n} (\eta_i - \bar{\eta}_{.})(e_i - \bar{e}_{.}) + c^2 \sum_{i=1}^{n} (e_i - \bar{e}_{.})^2 = z_1 + z_2 c + z_3 c^2$$

where $z_1$, $z_2$ and $z_3$ are computable constants for given simulated $(\eta_i, e_i)$.

(A4)

$$R^{2*} = 1 - SS_e^*/SS_y^* \Leftrightarrow$$

$$0 = SS_e^*/SS_y^* - 1 + R^{2*} \Leftrightarrow$$

$$0 = SS_e^* + (R^{2*} - 1)SS_y^* = c^2 SS_e + (R^{2*} - 1)(z_1 + z_2 c + z_3 c^2) = Ac^2 + Bc + C$$

This is a quadratic equation in $c$, which can be solved for $c$ by standard procedures. There are two distinct solutions but they result in errors that only differ in the sign, and so either solution can be chosen.

SAS-code to perform this simulation can be found in the supplementary materials of this paper.





 **Tables**

Table 1: Annual sediment load estimates (in Mg per year) for the two years of the study directly estimated without bootstrapping, and load estimates with 95% confidence interval limits and interval widths (difference between upper and lower limit) for the three different bootstrap methods: the full method shown in Figure 1, the method without modelled error (i.e. leaving out Step 3 in Figure 1) and the method without bootstrapping discharge (i.e. leaving out Step 1 in Figure 1). (n.a. = not applicable)


| | Error source | | 2010 | | | | 2011 | | | |
|---|---|---|---|---|---|---|---|---|---|---|
| Method | Autocorrelation | Q-equation | Estimate | Lower | Upper | Width | Estimate | Lower | Upper | Width |
| | | | Mg a$^{-1}$ | | | | Mg a$^{-1}$ | | | |
| Direct estimate | | | 5607 | n.a. | n.a. | n.a. | 4997 | n.a. | n.a. | n.a. |
| Full bootstrap method | ✓ | ✓ | 6262 | 4331 | 12267 | 7936 | 5543 | 3593 | 8975 | 5383 |
| Bootstrap without modeled error | | ✓ | 6575 | 4372 | 14586 | 10214 | 5839 | 3713 | 10410 | 6697 |
| Bootstrap without discharge | ✓ | | 5944 | 4203 | 11649 | 7446 | 5413 | 3521 | 8394 | 4876 |




**Figures**

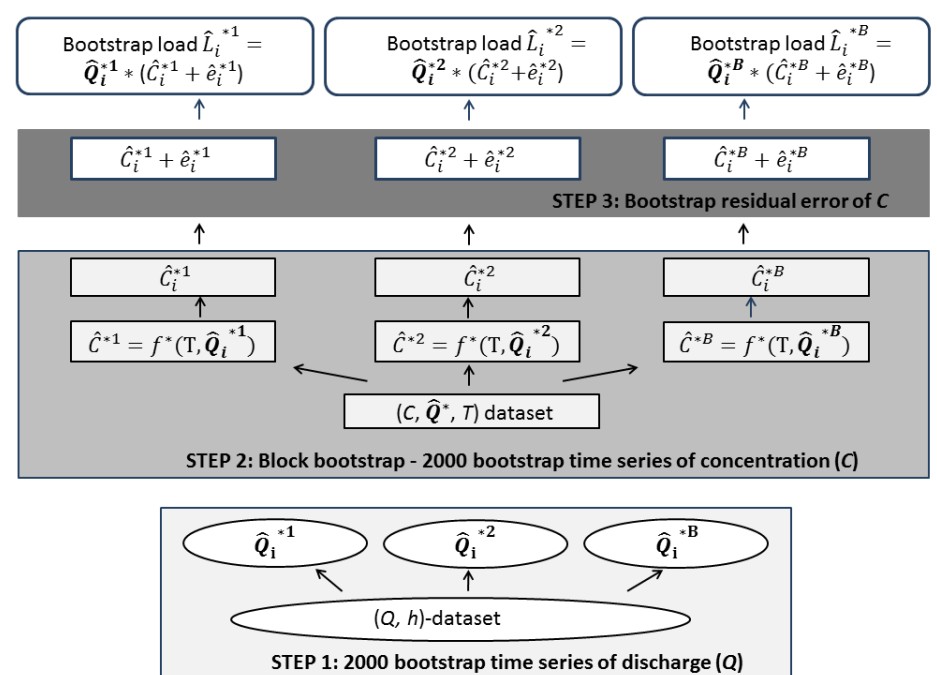

**Figure 1: Flow chart showing the three-step bootstrap mechanism**



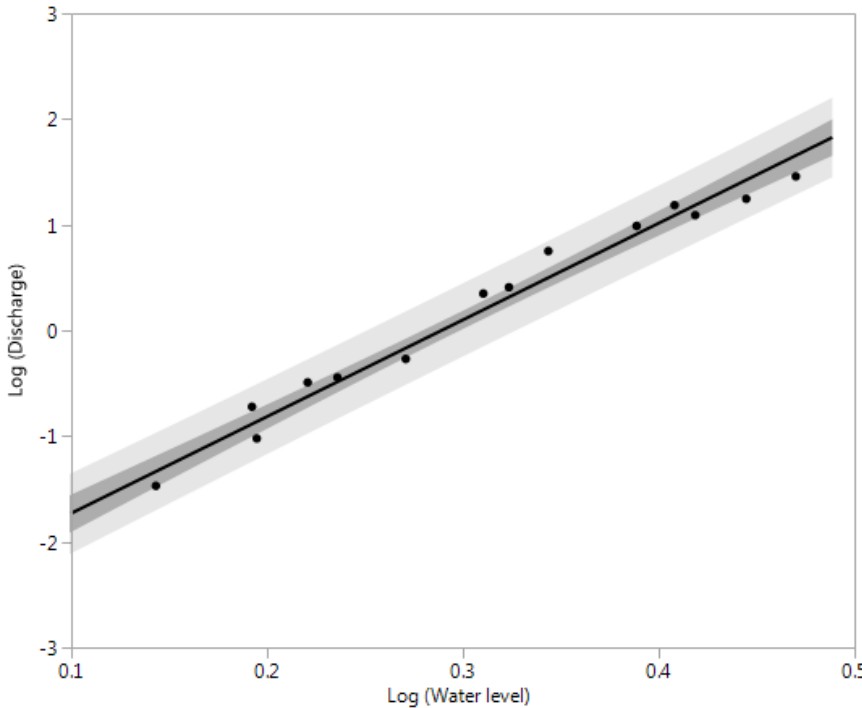

**Figure 2: Discharge rating curve plotted on the log-transformed scale showing 95% confidence interval for the**
**regression line (dark grey) and for new predictions (light grey). Stage-discharge rating curve: log(discharge) =**
**(9.0819*log(water level)) - 2.6423 (n=15, R$^2$=0.98)**





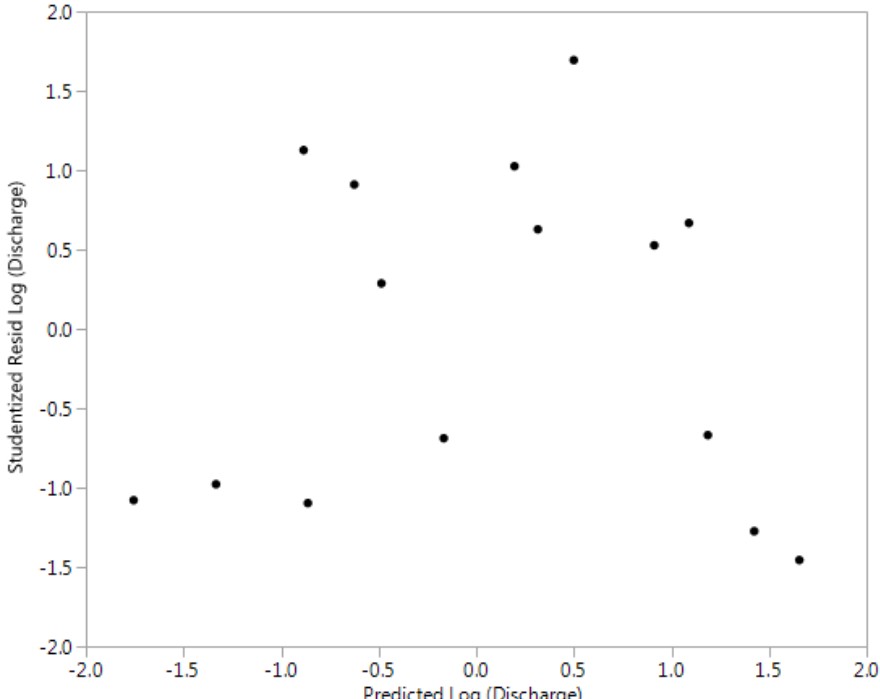

**Figure 3: Residual plot for the discharge rating curve, showing studentized residuals versus the predicted discharge**
**(on the log-transformed scale).**




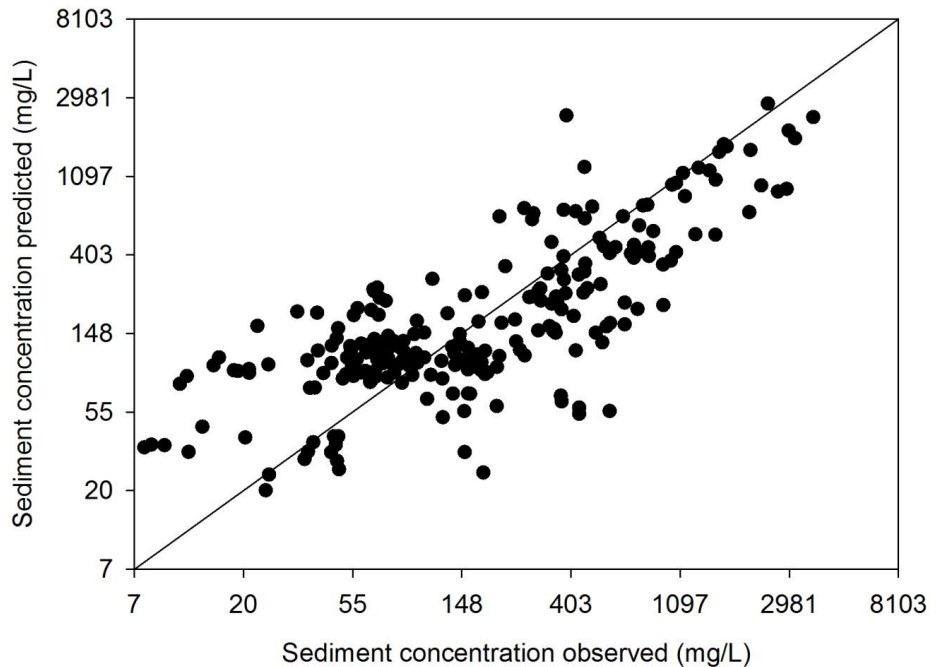

**Figure 4: Observed versus predicted values of the sediment rating curve. Predictions are from the linear mixed model with turbidity and discharge as quantitative predictor variables, and after five-fold cross validation (n=228, $r^2$=0.56). Axes are on the log-transformed scale while tick labels show values on the original scale.**






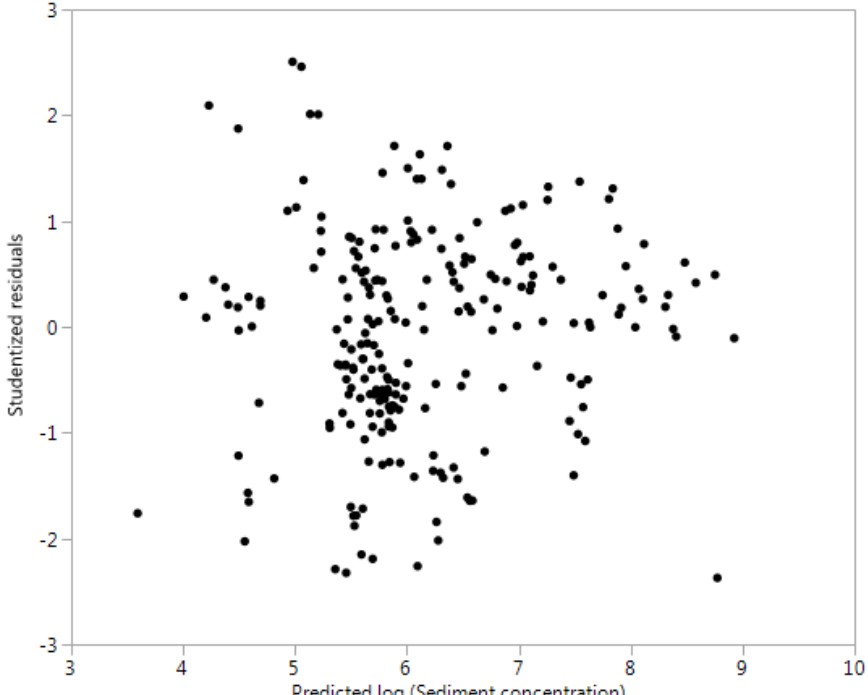

**Figure 5: Residual plot of the sediment concentration prediction model, studentized residuals versus the predicted sediment concentration (on the log-transformed scale).**



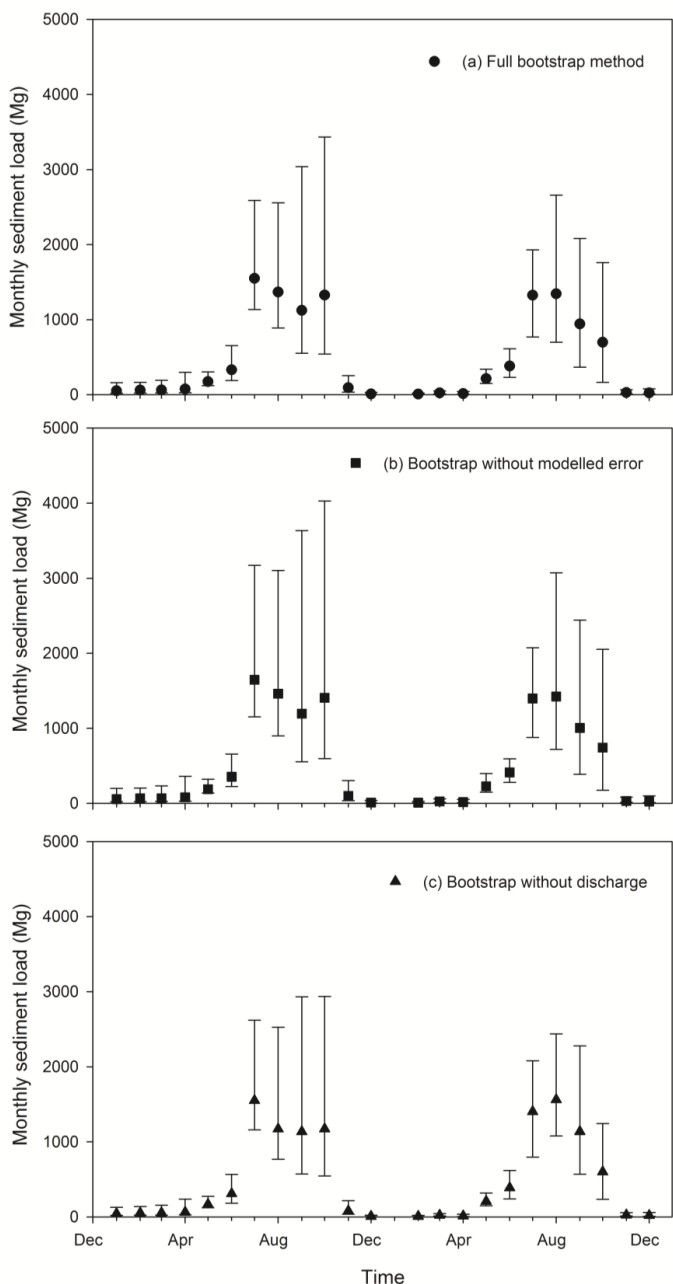


**Figure 6: Monthly sediment load estimates (in Mg per year) for the two years of the study with 95% confidence interval limits for the three different bootstrap methods: (a) the full method shown in Figure 1, (b) the method without modelled error (i.e. leaving out Step 3 in Figure 1) and (c) the method without bootstrapping discharge (i.e. leaving out Step 1 in Figure 1). In January 2011, discharge was zero; therefore no sediment load was transported**

**during this month.**





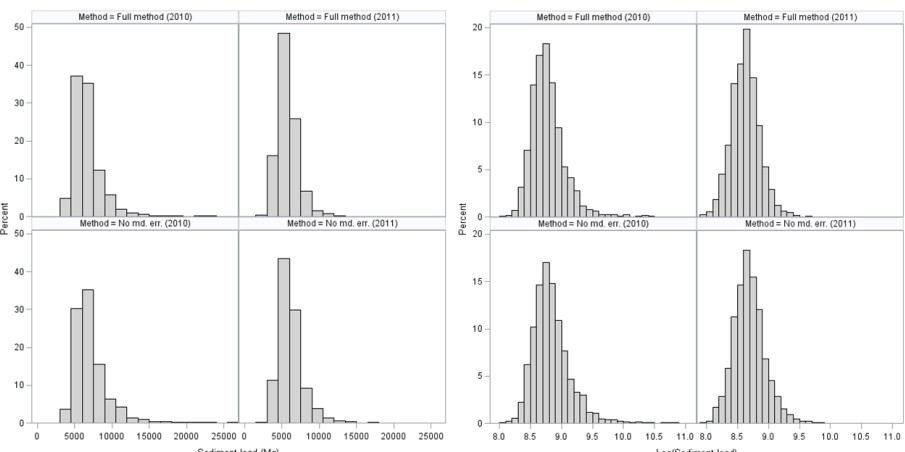

Figure 7: Histograms of bootstrap load estimates on the original scale (left) and the log-scale (right) for two study years and for two bootstrap methods: the full method with modeling the autocorrelated error ("Full method", top), and without modeling the error ("No md. err.", bottom)






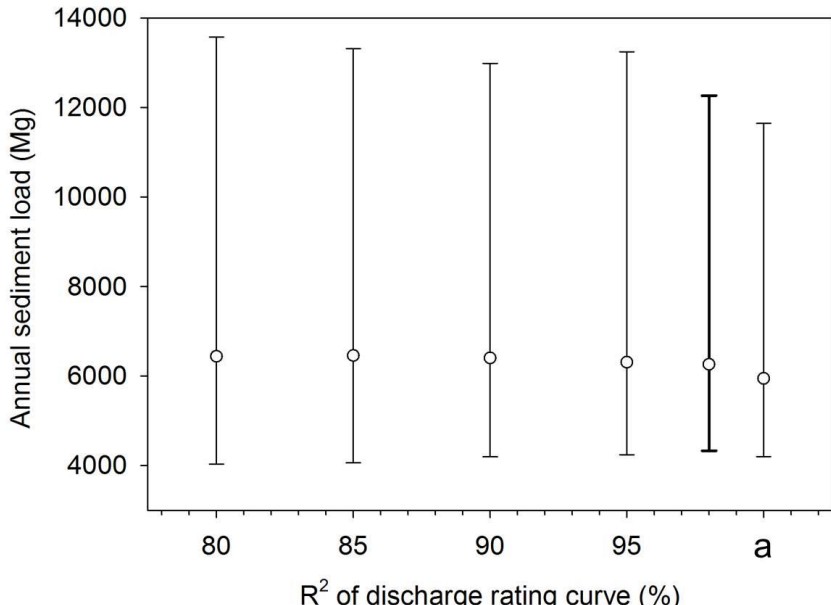

**Figure 8: Change in median and 95% confidence intervals for the sediment load estimate of 2010 (in Mg) when decreasing the coefficient of determination of the discharge rating curve. The bold line indicates the CI width of the real (Q,h) dataset. The letter "a" corresponds to not bootstrapping the (Q,h) pairs.**
