# Peer review of "Quantifying uncertainty on sediment loads using bootstrap confidence intervals"

_Hydrology and Earth System Sciences, 2016_

## Referee Comment (RC1) · T. Kumke (Referee) · 15 Aug 2016

T. Kumke (Referee)

thomas.kumke@ucb.com

Overall, this is a very well written paper on an important topic. The modelling approach using bootstrap estimates is an important approach and the authors nicely show the strength of the bootstrap. Here are some minor comments for consideration: (i) Introduction: the introduction seem to be quite exhaustive, for example there is a very long introduction on uncertainties, this could be surely reduced. On the other hand, serial correlation is an important aspect of the modelling approach, this has been hardly mentioned. Although, specific aims of the paper were introduced, I would strongly recommend that the authors state why their approach is very important for the estimation of loads. (ii) Bootstrap: I feel that the methodological aspects on the bootstrap could be reduced in length. I am sure that most readers are familiar with the basic principles. I have a few issues with some of the used language, for example l. 215: no clear winner? I am not sure whether there are winners or losers in a scientific context. Perhaps rephrasing helps here, eg, it remains unclear which of those specialized methods......
(iii) Results: The results of the modelling are nicely summarized. However, a couple of questions remains open. The effect of transformations should be evaluated according to the introduction, however, only the log-transformation (and back-transformation) was analyzed. What is the impact of other transformations on the CIs (ie, transformations with simple backtransforms)? 2000 bootstrap cycles were selected, however, it might be of interest, especially for readers not so familiar with the bootstrap, to explore the effect of the number of cycles on the estimates and CIs. The authors nicely explained the importance of including serial correlation, but in fact, it was only considered a first order autocorrelation. Did the authors explore at least a second order autocorrelation? Finally, did the authors consider to compare the bootstrap results with results of different complex models, eg GAM?

―――――――――――――――――――

---

## Referee Comment (RC2) · Anonymous Referee #2 · 20 Sep 2016

This paper is a very good contribution to the literature about load estimation and the uncertainty of load estimates. The consideration of the relative role of discharge uncertainty and concentration versus discharge uncertainty is a valuable contribution.

I do have concerns about the quality of the regression relationships that were used in the analysis. See my comments on figures 3 and 4. Also, they need to be clear about how they view year-to-year variability. Do they consider each year to be a separate population or are they each a different sample from the same population. Are estimates from the two years done separately? There are two schools of thought about how concentration prediction models should be built: using just data from the year of interest or using data from many years, with some consideration for the possibility that there may be a temporal trend in that relationship. They should be explicit about this issue.

[Figure]

line 222-229. It is not clear why the base-flow samples should be considered to be independent. My own experience is that they are not (I am thinking here about the residuals from a concentration versus discharge relationship). They used a first order autoregressive process but it seems to me that the process may be more complex than that, with memory that lasts for many days duration.

line 302. It appears to me that the method of removing transformation bias is similar to the approach called the "smearing estimate" proposed by Duan. I realize that Duan's smearing estimator is mentioned later in the paper (line 472) but I think it needs to be mentioned here as well. Furthermore, the question of whether the residuals are homoscedastic needs to be considered. If it is not, then this approach becomes problematic.

Figure 3. A sample size of 15 is very small for constructing a rating curve. It is disturbing that for the lowest two and highest three predicted values the residuals are all negative and fairly substantially so. My reaction to this plot is that there is some significant lack of fit to the proposed rating curve model. Even with this small number of observations, perhaps a higher order or non-linear model should have been considered.

Figure 4 is even more disturbing in terms of fit. The observations should be on the y-axis so that the residuals can be visualized as the vertical distance from the 1:1 line. For virtually every observed value greater than about 600 mg/L the residuals were all positive or only very slightly negative. Conversely, for the vast majority of the observed values below about 100 mg/L the residuals were almost all negative and in some cases the predictions were as much as 10 times greater than the observed. This is a very flawed model to be used as the basis for this experiment. A study of errors needs to start with a fitted model that does not have such a high degree of bias.

485-489. The issue is not whether concentrations or loads are log-normally distributed. The issue is the normality of the residuals from the fitted model. This is a common error

in analysis of such data sets. The adequacy of the estimation method should be based on the distribution properties of the residuals.

500-518. These points about overly complex models are very good. This is an important concern and I'm glad the authors emphasize it here.

―――――――――――――――

---

## Author Comment (AC1) · 18 Oct 2016

*Original comments in italic*
Responses in non-italic

*Overall, this is a very well written paper on an important topic. The modelling approach using bootstrap estimates is an important approach and the authors nicely show the strength of the bootstrap.*

We thank reviewer 1 for his positive evaluation of our paper. Specific comments are addressed below:

*Here are some minor comments for consideration: (i) Introduction: the introduction seems to be quite exhaustive, for example there is a verylong introduction on uncertainties, this could be surely reduced. On the other hand, serialcorrelation is an important aspect of the modelling approach, this has been hardlymentioned. Although, specific aims of the paper were introduced, I would strongly recommendthat the authors state why their approach is very important for the estimationof loads.*

The introduction section on sources of uncertainty for sediment and discharge rating curves has been shortened.
Furthermore, the aspect of serial correlation in sediment rating curves in literature has been given more emphasis in the introduction: "… For sediment concentration, however, flow-proportional sampling is often performed to obtain samples at the highest concentrations. Those observations are usually taken closely together during storms and thus most likely are not independent in time (Slaets et al., 2014). Linear mixed models that account for serial correlation provide an alternative to least squares regression to establish a sediment rating curve for this type of data. Lessels and Bishop (2013) similarly found that the inclusion of a temporal autocorrelation component improved the accuracy and decreased the bias in predictions of total phosphorus and nitrogen river loads.  If there is serial correlation in the sediment data, it is necessary to use an adjusted version of the bootstrap that retains the serial correlation in the data intact (Lahiri, 2003). Such methods have already been explored in hydrology in relation to the discharge rating curve: Ebtehajet al. (2010) and Selle and Hannah (2010) uses block bootstrap methods to assess uncertainty on and improve robustness of model parameter estimates for discharge prediction."
Additionally to the specific aims, we have added a section on the importance of our approach for load estimation: "Combining these aspects, the proposed method provides a means to assess uncertainty on any type of constituent load which was calculated from continuous constituent concentration and discharge predictions estimated with regression-type methods. The approach thus allows load estimates to be reported with an uncertainty assessment, rather than as a point estimate alone, making them informative to end users and decision makers."

*(ii) Bootstrap: I feel that the methodological aspects on the bootstrap couldbe reduced in length. I am sure that most readers are familiar with the basic principles.*

We have shortened the methodological section on the bootstrap and refer the reader to Efron and Tibshirani (1993) for further details.

*I have a few issues with some of the used language, for example l. 215: no clear winner? I am not sure whether there are winners or losers in a scientific context. Perhaps rephrasing helps here, eg, it remains unclear which of those specialized methods......*

The sentence has been rephrased to: "Among these specialized methods, no preferred method has emerged from literature. Furthermore, many of these methods require a vast set of decisions with regards to for example the block size for which no general recommendation exists. As a consequence, results from different methods are not straightforward to compare."

*(iii) Results: The results of the modelling are nicely summarized. However, a couple of questions remains open. The effect of transformations should be evaluated according to the introduction, however, only the log-transformation (and back-transformation) was analyzed. What is the impact of other transformations on the CIs (ie, transformations with simple backtransforms)?*

The choice for a specific data transformation is driven by the need to stabilize the residual error variance. In our case, the log transformation was the one that was successful in doing so. The log transformation is very commonly used in load studies, and the most frequently used alternative data transformations in load estimation are typically other power transformations such as 1/Y, log(Y), square root, cube root, and fourth root, all of which are Box Cox transformations. These were tested but were not successful in obtaining normality and homoscedasticity of residuals, as is shown here in the diagnostic plots for the square root transformation:

[Figure]

As the diagnostic plots show that other commonly used transformations in the Box-Cox family are less appropriate for our dataset, we would rather not to present results for another transformation in the Box-Cox family. The exploration of alternative transformations has now been added to the Material and Methods section of the paper: "Other transformations, such as the square root, were inspected using residual plots and were found to be unsuitable for meeting the assumptions of normality and homoscedasticity."

Alternatives to Box Cox transformations such as the exponential family of Manly (for non-positive data) are also available, but far less frequently seen in load estimation: the main reason for data transformation in load studies is heterogeneity of variance, and depending on the strength of this heteroscedasticity, some form of the power Box-Cox family is usually successful in stabilizing the variance.

In order to make our discussion of the implications of the log transformation more generally applicable in the case of other transformations, we have added the following statements to show the similarities for the whole Box-Cox power family of transformations to the discussion section: "The most commonly used data transformations in load estimation are typically other members of the Box Cox power family, such as 1/Y, square root, cube root, and fourth root. Transformations in this family are usually required where the original data exhibit pronounced skewness and heteroscedasticity, which is generally the case in load studies. Therefore for all transformations in the Box Cox family, naïve back-transformation of estimates would similarly result in biased estimates of means on the original scale, as was illustrated with the log-transformation in our dataset." To further emphasize the importance of the choice of the transformation, which needs to be appropriate to the data at hand, the following text has been added after explaining the various correction options: "Regardless of the chosen correction factor, it is important that homoscedasticity after the transformation is confirmed by visually inspecting the diagnostic plots, as was done in the case of this dataset."

*2000 bootstrap cycles were selected, however, it might be of interest, especially for readers not so familiar with the bootstrap, to explore the effect of the number of cycles on the estimates and CIs.*

We thank the reviewer for this interesting suggestion. In order to assess the effect of the number of bootstrap replicates, we have re-run the load estimation for one year with 500, 1000 and 1500 bootstrap cycles. The resulting histograms are shown in Figure 8 and their implications discussed in the results section on the required number of bootstrap replicates. As the crucial point is the smoothness of the histograms in order to have reliable bootstrap estimates, especially in the tails as we are looking at confidence intervals via the percentile method, there is no straightforward relationship with number of bootstrap iterations and wider (or narrower) CI's, or lower or higher estimates. Rather, the lack of smoothness especially in the tails makes the estimates unreliable, which we have clarified by adding the following text: "Before looking at the bootstrap confidence intervals, the histograms of the bootstrap load estimates were evaluated (Figure 7). The histogram of the 2000 bootstrap estimates looked reasonably smooth, so we concluded that sample size was adequate for the percentile bootstrap. When reducing the number of bootstrap replicates (Figure 8), the change in smoothness, especially in the right tail, becomes visible. Tail smoothness of the empirical distribution is a requirement when using the percentile method to obtain confidence intervals (Efron and Tibshirani, 1993). At 500 bootstrap replicates, the centre of the distribution displays lack of smoothness as well, thus not only affecting the confidence interval estimates, but the load estimates as well."

*The authors nicely explainedthe importance of including serial correlation, but in fact, it was only considered a firstorder autocorrelation. Did the authors explore at least a second order autocorrelation?*

In Slaets et al. (2014), where the same dataset was used, we explored the use of several alternative variance-covariance structures to model the serial correlation. The Akaike Information Criterion (AIC) was used to compare various candidate models and a spatial power structure with time as the coordinate showed the best fit based on this criterion. In the bootstrap iterations, however, the spatial power structure caused convergence issues in a large number of the bootstrap replicates which would result in biased bootstrap estimates, and therefore we switched back to the AR(1) structure. The power model is an extension of the AR(1) model to accommodate for unequally spaced observations, and both structures are essentially different parameterizations of the same model (Piepho et al., 2015). In the spatial power model, the autocorrelation decays as a function of the distance between observations (in this case, the distance in time). The difference in AIC between the AR(1) and spatial power model was 4 points, indicating that while the spatial power model is most likely the best performing model, there is still considerable support for the AR(1) model (Burnham and Anderson, 2002). Unfortunately the AR(2) structure is not available in the Mixed procedure of SAS, and if it were available, we would expect to encounter convergence issues in running the bootstrap iterations. As the AIC of the AR(1) was comparatively close to that of the spatial power model, however, the AR(1) structure was an adequate approximation of the serial correlation structure in the data.

We have added the discussion of the spatial power model to the methods section: "In a previous model published in Slaets et al. (2014), we explored the use of several alternative variance-covariance structures to model the serial correlation. The selected spatial power model unfortunately caused non-convergence for a large number of the bootstrap replicates when using it for bootstrap load estimates, and therefore the AR(1) structure was implemented as it did not have convergence issues. The difference in AIC between the AR(1) and spatial power model was 4 points. Therefore the spatial power model is most likely the best performing model, but there is still considerable support for the AR(1) model (Burnham and Anderson, 2002)."

*Finally, did the authors consider to compare the bootstrap results with results of different complex models, eg GAM?*

The most common GAMS can be estimated using maximum likelihood methods. For more complex GAMS, we are not aware of any procedures in SAS that implement ML algorithms that can also fit random effects and serial correlation. An alternative is computing a profile likelihood but we believe that least squares and maximum likelihood methods are by far the most commonly used methods to establish sediment rating curves. One possible alternative would be to use the Glimmix procedure to fit B-splines, which are very similar to GAMs, to explore nonlinearities as the Glimmix procedure can model random effects. With the level of noise in the sediment rating curve, however, there is a danger of overfitting unless very clear irregular nonlinear shapes are seen in the data. Therefore we consider comparison to generalized additive models to be outside the scope of our paper, though we refer to their potential for further exploration of load estimation uncertainty in the conclusions: "Reporting uncertainty is especially important when water quality models are complex. There has been a great increase in the use of more complex predictive methods for water quality, for example the use of Artificial Neural Networks, Random Forests or Generalized Additive Models (Berk, 2008). The advent of these methods makes the consistent reporting of measures of uncertainty even more essential: the more complex a model is, the more prone it is to overfitting (Burnham and Anderson, 2002), as was demonstrated by the inflated confidence intervals when adding predictor variables to the sediment concentration model."

---

## Author Comment (AC2) · 18 Oct 2016

*Original comments in italic*
Responses in non-italic

*This paper is a very good contribution to the literature about load estimation and the uncertainty of load estimates. The consideration of the relative role of discharge uncertainty and concentration versus discharge uncertainty is a valuable contribution.*

We thank reviewer 2 for his positive evaluation of our paper. Specific comments are addressed below:

*I do have concerns about the quality of the regression relationships that were used in the analysis. See my comments on figures 3 and 4. Also, they need to be clear abouthow they view year-to-year variability. Do they consider each year to be a separate population or are they each a different sample from the same population. Are estimates from the two years done separately? There are two schools of thought about how concentration prediction models should be built: using just data from the year of interestor using data from many years, with some consideration for the possibility that there may be a temporal trend in that relationship. They should be explicit about this issue.*

We consider the two-year period to be one population and use all samples from the two year study period to build the prediction models. Both years are thus predicted from the same model, the parameters of which are estimated from the same data. This information has been added now to the Materials and Methods section as follows: "All samples from the two year study period were used to build the concentration prediction model, and load estimates from both years are thus predicted from the same model with the same parameter estimates."

*line 222-229. It is not clear why the base-flow samples should be considered to be independent. My own experience is that they are not (I am thinking here about the residuals from a concentration versus discharge relationship). They used a first order autoregressive process but it seems to me that the process may be more complex than that, with memory that lasts for many days duration.*

In a previous model published in Slaets et al. (2014), we explored the use of several alternative variance-covariance structures to model the serial correlation. The Akaike Information Criterion (AIC) was used to compare various candidate models and a spatial power structure with time as the coordinate showed the best fit based on this criterion. The variance-covariance parameter estimates from this model showed that the autocorrelation becomes nearly zero for samples taken more than 80 min apart– which coincides with the average duration of rainfall events, and was the basis for the independence assumption for the base-flow samples. The reviewer rightly points out that in many natural catchments, base-flow samples are not necessarily independent. The different result for our dataset might be attributable to the irrigation management present in the paddy-containing watershed, which disturbs natural hydrographs and changes the memory effect.
The selected spatial power model unfortunately caused non-convergence for a large number of the bootstrap replicates when using it for bootstrap load estimates, and therefore the AR(1) structure was implemented as it did not have convergence issues. The difference in AIC between the AR(1)

and spatial power model was 4 points. Therefore the spatial power model is most likely the best performing model, but there is still considerable support for the AR(1) model (Burnham and Anderson, 2002).

We have clarified the independence assumption for base-flow samples in the manuscript: "In a previous model published in Slaets et al. (2014), we explored the use of several alternative variance-covariance structures to model the serial correlation. The selected spatial power model unfortunately caused non-convergence for a large number of the bootstrap replicates when using it for bootstrap load estimates, and therefore the AR(1) structure was implemented as it did not have convergence issues. The difference in AIC between the AR(1) and spatial power model was 4 points. Therefore the spatial power model is most likely the best performing model, but there is still considerable support for the AR(1) model (Burnham and Anderson, 2002). The spatial power structure with time as the coordinate showed that the autocorrelation becomes nearly zero for samples taken more than 80 min apart– which coincides with the average duration of rainfall events. Therefore the base-flow samples were considered to be independent."

*line 302. It appears to me that the method of removing transformation bias is similar to the approach called the "smearing estimate" proposed by Duan. I realize that Duan's smearing estimator is mentioned later in the paper (line 472) but I think it needs tobe mentioned here as well. Furthermore, the question of whether the residuals are homoscedastic needs to be considered. If it is not, then this approach becomes problematic.*

The two methods used for removing transformation bias in our approach are both parametric (adding half the residual error variance or simulating an AR(1) term). Duan's smearing estimator would be a third option and a non-parametric alternative, as it is a correction where the sample average of the exponentiated residuals from the model is used as the correction factor. Duan's smearing estimator assumes iid errors. As such, the procedure is not a suitable alternative in this study as serial correlation in the data is present. We have added these clarifications to the Methods section on data transformations: "With nonlinear data transformations (the log-transformation and the Box-Cox transformation being prime examples), predicted means cannot be naively back-transformed and interpreted as means on the original scale. Correction factors can be applied that compensate for the underestimation of SSC that arises from doing the predictions on the transformed scale. A commonly used non-parametric correction factor is Duan's smearing estimator (Duan, 1983), where the sample average of the exponentiated residuals from the model is used as the correction factor. Duan's smearing estimator assumes independent and identically distributed errors, however, and is therefore also not a suitable alternative when serial correlation in the data is present. Alternatively, as pointed out by Rustomji and Wilkinson (2008), adding the modeled residual error removes the need to apply a correction factor and is therefore the recommended approach."

The reviewer correctly points out that, irrespective of the chosen correction factor, it must be confirmed that the transformation was successful in stabilizing the variance. This point has been emphasized after explaining the various correction options: "Regardless of the chosen correction factor, it is important that homoscedasticity after the transformation is confirmed by visually inspecting the diagnostic plots, as was done in the case of this dataset."

*Figure 3. A sample size of 15 is very small for constructing a rating curve. It is disturbing that for the lowest two and highest three predicted values the residuals are all negative and fairly substantially so. My reaction to this plot is that there is some significant lack of fit to the proposed rating curve model. Even with this small number of observations, perhaps a higher order or non-linear model should have been considered.*

As the reviewer points out, the lowest two and highest three studentized residuals are negative, however they are not larger than two (and in fact smaller than 1.5). Furthermore, as can be seen from Figure 2, the 95% confidence intervals for the regression line as well as for new predictions are narrow and show stability over the whole range of flows. Nonlinearity was explored by adding a quadratic term for the log-transformed water level, but the adjusted $R^2$ changed only marginally (from 0.978 to 0.985), therefore we opted for the more parsimonious linear model.

The reviewer makes a good point that there is an effect of sample size on the bootstrap confidence intervals: the sample size in this study is small, and a larger sample size could thus result in a lower uncertainty on the load estimates of the discharge rating curve. Several aspects come into consideration when assessing sample size in the context of bootstrapping a stage-discharge rating curve. In very small sample sizes, the bootstrap could fail due to the set of possible bootstrap samples not being rich enough. For sample sizes as small as 8 and up, however, the number of distinct bootstrap samples gets large enough very quickly to result in consistent estimates (Chernick, 2011; Hall, 2013). Our sample of fifteen is thus large enough to bootstrap. As for the number of observations required to obtain a reliable discharge rating curve: no fixed sample size recommendation exists, as the accuracy of the discharge rating curve is determined not solely by the sample size, but also the stability of the river bed, shape of the cross-sectional profile, the range of discharges, and foremost the spread of the samples, as all ranges of flow need to be covered. The studied catchment has a small range of discharges (i.e. from 0.15 $m^3$/s to 4.30 $m^3$/s corresponding to minimum water levels of 1.1 m and maximum water levels of 1.6 m), a stable river bed and a short duration of the study (two years). And importantly, the dataset has a good spread over the ranges of discharge (Figure 2) covering the lowest and highest water levels measured throughout the 2010-2011 period for that specific location. These points regarding the effect of sample size and limitation of our dataset have now been added to the discussion: "Even though the discharge rating curve has a high accuracy, an estimate of Q is used as a predictor variable for concentration, and the concentration then gets multiplied with the estimate of Q, and so the effect is not as small as one would expect based on the $R^2$ of both rating curves. It is possible that the sample size of the discharge rating curve, which is relatively small (n=15) plays a role here, as a bootstrap iteration that does not contain the largest discharge values would result in a wider confidence interval for the estimated load."

*Figure 4 is even more disturbing in terms of fit. The observations should be on the y-axis so that the residuals can be visualized as the vertical distance from the 1:1 line. For virtually every observed value greater than about 600 mg/L the residuals were all positive or only very slightly negative. Conversely, for the vast majority of the observed values below about 100 mg/L the residuals were almost all negative and in some cases the predictions were as much as 10 times greater than the observed. This is a very flawed model to be used as the basis for this experiment. A study of errors needs to start with a fitted model that does not have such a high degree of bias.*

Thank you for this good suggestion, we have now changed the figure to display the observed values on the Y-axis as the reviewer suggested. Figure 4 displays the observed versus the predicted values of the sediment rating curve after five-fold cross validation, and as such, it does not show model residuals or allows for the checking of model assumptions. Residuals of the sediment concentration model can be seen in Figure 5 plotted against the predicted values. As we use a linear mixed model with two quantitative predictors (discharge and turbidity) for our concentration predictions, we are not able to show a plotted regression equation; and plotting observed versus predicted values as we do in Figure 4 is an alternative way to visualize predictive accuracy of the model. As we show the predicted values after cross validation, this plot is a stronger measure of accuracy than a classic regression line. Figure 4 illustrates, as the reviewer correctly points out, that the concentration model for new predictions tends to over predict low concentrations and under predict high concentrations. This tendency of regression towards the mean is not a flaw of the model, but

typically seen when models are fitted to very noisy data and is also well documented in erosion studies (Nearing, 1998). We thank the reviewer for bringing up this important trend in Figure 4 and have added the discussion of this phenomenon in the text: "The sediment rating curve tends to over predict low concentrations and under predict high concentrations for new data, as is visible in Figure 4. This tendency of regression towards the mean is typically seen when models are fitted to very noisy data, and is also well documented in erosion studies (Nearing, 1998)."

While we therefore do not see model bias confirmed in the residual plots, the trend in Figure 4 demonstrates that the predictive power of our model has limitations, as is also clear from the reported Pearson's $r^2$ between observed and predicted values after cross validation of 0.56. For datasets similar to our own, in fragmented landscapes, with heterogeneous terrain, soils, geology and land use, a model that explains half or more of the variation after validation would be difficult to improve without overfitting with a more complex model. That being said, in other catchments with less heterogeneity where a much more accurate sediment rating curve can be obtained, the resulting load estimates will be more accurate and it is a limitation of our dataset that we cannot assess confidence interval width for such a scenario. Therefore we have addressed these limitations in the discussion: "The accuracy of the sediment rating curve in this study (Pearson's $r^2$=0.56 after cross validation) is reasonable for catchments with large heterogeneity in relief, land use, soil types and rainfall event characteristics. In more homogeneous settings, however, much more accurate sediment rating curves have been obtained, which can be expected to result in more narrow confidence intervals on their resulting load estimates."

*485-489. The issue is not whether concentrations or loads are log-normally distributed. The issue is the normality of the residuals from the fitted model. This is a common error in analysis of such data sets. The adequacy of the estimation method should be based on the distribution properties of the residuals.*

The reviewer correctly points out that there is no need for concentrations or loads to be log-normally distributed, but rather model assumptions need to be checked on residuals in the case of regression type models. The point we were trying to make here was regarding the use of the delta-method as an alternative to obtain an estimate of the load variance. In this approach, log-normality of the loads was required, and therefore in the case of our dataset, the method would not have yielded valid results. To clarify this for the reader, the corresponding section has been changed as follows: "Regarding the data transformation, while the sediment concentration was log-normally distributed, the log-transformed load estimates were not normally distributed (Figure 7, right panel). This non-log-normality of our loads does not affect the viability of the bootstrap approach, as regression type methods do not require the concentration or load data but rather normality of the residuals. It does, however, limit the applicability of methods that use the log-normality assumption of the load to estimate a variance for the load, as was done for example by Wang et al. (2011) in an approach that used the delta-method as an alternative way to assess uncertainty on annual sediment load estimates."

*500-518. These points about overly complex models are very good. This is an important concern and I'm glad the authors emphasize it here.*
We thank the reviewer for his positive evaluation of this point.

**References**

Burnham, K. P., and Anderson, D. R.: Model Selection and Inference: A Practical Information-Theoretic Approach, Springer-Verlag GmbH, New York, 1998.

Chernick, M. R.: Bootstrap methods: A guide for practitioners and researchers, John Wiley & Sons, 2011.

Duan, N.: Smearing estimate: a nonparametric retransformation method, Journal of the American Statistical Association, 78, 605-610, 1983.

Hall, P.: The bootstrap and Edgeworth expansion, Springer Science & Business Media, 2013.

Nearing, M. A.: Why soil erosion models over-predict small soil losses and under-predict large soil losses, Catena, 32, 15-22, 1998.

Rustomji, P., and Wilkinson, S. N.: Applying bootstrap resampling to quantify uncertainty in fluvial suspended sediment loads estimated using rating curves, Water Resources Research, 44, W09435, 2008.

Slaets, J. I. F., Schmitter, P., Hilger, T., Lamers, M., Piepho, H. P., Vien, T. D., and Cadisch, G.: A turbidity-based method to continuously monitor sediment, carbon and nitrogen flows in mountainous watersheds, Journal of Hydrology, 513, 45-57, 10.1016/j.jhydrol.2014.03.034, 2014.

Wang, Y. G., Kuhnert, P., and Henderson, B.: Load estimation with uncertainties from opportunistic sampling data - A semiparametric approach, Journal of Hydrology, 396, 148-157, 2011.